# KNOW-THE-ROPES: ALGORITHMIC BLUEPRINTS FOR RELIABLE LLM MULTI-AGENT SYSTEM DESIGN

## ABSTRACT

Single-agent LLMs face finite context and role overload, while unstructured multi-agent designs can introduce ambiguous roles and coordination overhead. We therefore introduce Know-The-Ropes (KtR), a practical methodology for projecting algorithmic priors and heuristics into typed, controller-mediated multi-agent blueprints for decomposable tasks. KtR follows a multi-step process—identify bottlenecks, refine decomposition, apply minimal augmentation (chain-of-thought, self-check, or light fine-tuning), and verify via contracts. In two case studies, including Knapsack (3–8 items) and Task Assignment (6–15 jobs), we find that KtR by low-effort LLMs can show notable end-to-end accuracy gains over single-agent zero-shot baselines. With three GPT-4o-mini agents, accuracy on size-5 Knapsack instances rises from 3% to 95% after addressing a single bottleneck agent. With six o3-mini agents, Task Assignment reaches 100% up to size 10 and $\geq 84\%$ on sizes 13–15, versus $\leq 11\%$ zero-shot. These results indicate benefits in our controlled setting; These results indicate benefits in our controlled setting; KtR complements scaling and prompt/program-of-thought techniques in building a reliable multi-agent system. An anonymous code base is available at this anonymous link

## 1 INTRODUCTION

Large language model (LLM) agents typically achieve strong performance in the domains for which they are trained and optimized (Thirunavukarasu et al., 2023; Kasneci et al., 2023; Wu et al., 2023b), yet their effectiveness degrades outside those boundaries. Finite context windows limit long-document reasoning, and no single agent can simultaneously address mathematics, coding, and planning problems (Liu et al., 2023; Gulati et al., 2024; Wu et al., 2025). Persistent challenges, including hallucinations, topical drift, and domain-specific failures, further constrain their reliability (Zhang et al., 2024; Xu et al., 2024). A natural direction is the division of labor by decomposing tasks across specialized agents that coordinate to produce a joint solution. Current frameworks such as Mixture-of-Agents suggest that, when properly orchestrated, a team of agents can outperform even its strongest individual member (Wang et al., 2024a).

Yet significant challenges remain. First, each task still demands a carefully crafted prompt, often requiring substantial manual effort (Li et al., 2025; Cui et al., 2025). While some multi-agent frameworks report encouraging results, once evaluation leakage and prompt overfitting are controlled, the apparent gains of naïve agent swarms collapse to single digits, and can even turn negative when tasks require more rounds of coordination (Pan et al., 2025; Zhu et al., 2025). Post-hoc analyses consistently reveal recurrent failure modalities: ill-posed task decompositions propagate ambiguity, imprecisely defined roles lead to redundancy or coverage gaps, and verification mechanisms are either predicated on brittle heuristics or become computationally prohibitive (Cao et al., 2025; Wang et al., 2024b). Furthermore, latency and cost tend to scale super-linearly with each round of interaction (Ye, 2025; Shu et al., 2024). These findings suggest that simply adding more "brains" does not guarantee progress. Robust and scalable improvements of multi-agent system (MAS) design demands a principled, systems-engineering approach–the gap our work aims to close.

To address this critical need, we introduce Know-The-Ropes (KtR), a framework that reframes MAS design as structured algorithm engineering. At its core, KtR employs a hierarchical task decomposition, recursively partitioning problems into sub-tasks until each primitive operation is

solvable by a base model, potentially augmented with minimal augmentation (e.g., self-check loops, Chain-of-Thought (CoT) (Wei et al., 2022), fine-tuning). Inter-agent communication is not conversational but is instead mediated by a code-based controller that enforces explicit, typed input/output contracts. This architectural choice ensures modularity and predictable information flow, preventing common pathologies such as context bloat and state overwrites. This design philosophy turns MAS construction into traditional algorithm engineering: identify bottlenecks, refine the decomposition, and apply the most cost-effective augmentation to underperforming agents, mirroring the optimization of classical computational graphs.

Our empirical study includes the following experiments: **(i) Knapsack Problem.** Starting with GPT-4o-mini, zero-shot accuracy on 3–8-item instances ranges from 60% to 0%. A naïve task-level fine-tune shows limited improvement. Under our KtR three-agent blueprint, accuracy improves to 95%–70% in our controlled setting after fine-tuning just the "trimmer" sub-task on 1200 worked examples. **(ii) Task-Assignment Problem (scalability case study).** With o3-mini we build a six-agent blueprint for problems of size 6–15. Decomposing a single weak agent into two finer leaves yields leaf accuracies of 100% and 97%, with the overall system achieving $\geq 84\%$ accuracy across all sizes in our evaluation protocol. Based on these findings, our contributions lie in three aspects.

- **A New Framework for MAS Design**. We present KtR, a practical framework for translating algorithmic priors into multi-agent blueprints with typed interfaces and local verification, applicable to decomposable tasks with known domain structure.
- **Empirical Validation**. Two illustrative demonstrations (Knapsack and Task-Assignment problems) indicate that targeted decomposition can improve end-to-end accuracy versus single-agent baselines in our settings, using modest models and minimal augmentation.
- **Practices for Principled Refinement**. We provide diagnostic tools (per-agent accuracy metrics, tractability checks) and minimal augmentation strategies (CoT, self-check, light fine-tuning) for blueprint refinement.

## 2 RELATED WORK

Multi-Agent Systems (MAS) have been widely employed to enhance the capabilities of LLMs to tackle complex tasks (Qiu et al., 2024; Yan et al., 2024; Ma et al., 2024; Lin et al., 2024; Hua et al., 2023; Yu et al., 2024). This is because MAS typically distribute tasks across agents that collaborate to achieve a common goal, thereby improving both efficiency and adaptability. Recent frameworks like CAMEL (Li et al., 2023) enable role-based cooperative dialogues by assigning agents distinct personas, while AutoGen (Wu et al., 2023a) and MetaGPT (Hong et al., 2023) orchestrate multi-role agent teams through structured conversation loops and predefined workflows. In math optimization, OR-LLM-Agent can translate natural-language problem descriptions into formal Gurobi models—achieving an 85% correct-solution rate on real-world benchmarks (Zhang & Luo, 2025).

However, studies show that simply scaling up to LLM-based MAS often yields only marginal gains over single-agent baselines (Pan et al., 2025). LLM agents still struggle with context management and consistency, meaning that elaborate multi-agent prompts can fail to realize the intended collaboration (Bo et al., 2024). A recent systematic audit of popular MAS frameworks has identified 14 distinct failure modes (Cemri et al., 2025), which can be grouped into three categories, including flawed design (e.g., ambiguous role definition), inter-agent misalignment (e.g., communication failures), and quality control (e.g., no reliable check mechanism).

To address these challenges, researchers have proposed multiple strategies to make LLM-based MAS more reliable (Zhu et al., 2025; Tran et al., 2025). A key strategy is improving the agent interaction structure (Zhu et al., 2025). For example, the AgentDropout framework proposes a dynamic agent-pruning strategy, which seeks to discard less critical actors during training (Wang et al., 2025). Another effective strategy is incorporating feedback and verification loops (Hong et al., 2023). A recent study shows that frameworks with role specialization and iterative feedback mechanisms can outperform those without these features (Anonymous, 2025). In addition, systematic evaluations suggest that the communication topology matters: a well-designed protocol between agents can significantly improve collective performance on complex tasks (Zhu et al., 2025).

To understand how KtR relates to existing approaches, Table 1 compares our framework with both MAS and prompting methods across three design dimensions KtR differs from frameworks that rely on emergent coordination (e.g., CAMEL) or developer orchestration (e.g., AutoGen) by imposing

algorithmic blueprints with typed interfaces and local pre/post. Compared with prompting methods (ReAct, ToT), KtR embeds domain–specific algorithmic knowledge directly into the multi-agent architecture. This positions KtR as a reliability complement to model scaling and prompt design for decomposable tasks.

Table 1: Comparison of MAS frameworks and prompting baselines.

| Method | Inductive Bias | Verification | Typed I/O |
|---|---|---|---|
| **KtR (Ours)** | Algorithmic blueprint | Local pre/post checks | JSON-schema |
| AutoGen (Wu et al., 2023a) | Dev. orchestration | Limited | Programmatic |
| MetaGPT (Hong et al., 2023) | SOPs & templates | Role-based QA | Structured |
| CAMEL (Li et al., 2023) | Emergent coordination | Minimal | Free-text |
| ReAct (Yao et al., 2023b) | Tool-use structure | Limited | Func. calls |
| ToT (Yao et al., 2023a) | Thought tree | Limited | Text-based |
| GoT ()besta2024graph | Graph reasoning | None | Text-based |
| DSPy (Khattab et al., 2024) | Programmatic prompting | Optimization | Signatures |
| PAL Gao et al. (2023) | Program synthesis | Code execution | Structured |

## 3 METHODOLOGY-KTR FRAMEWORK DESIGN

We propose the heuristic framework "Know the Ropes" (KtR). KtR offers a structured methodology for designing specialized MAS leveraging LLMs. This heuristic focuses on translating known, effective procedures or algorithms into a coherent multi-agent architecture. As presented in Figure 1, the core idea is to decompose a complex overall task into its fundamental computational stages. Each stage is then mapped to a well-formulated sub-task, designed to be tractable for an individual agent. These specialized agents are subsequently orchestrated to mirror the data/control flow of the original procedure, which can effectively embed problem-solving logic into the MAS. The following definitions formalize the components of this framework.

This approach is grounded in the No-Free-Lunch (NFL) theorem (Wolpert & Macready, 1997; Wolpert, 2021) (see Appendix B), which states that no algorithm performs universally better than others across all problem distributions. Rather than seeking a universal multi-agent orchestration strategy, KtR operationalizes the NFL insight by injecting domain-specific inductive bias through algorithmic blueprints. By decomposing tasks according to their inherent algorithmic structure, we concentrate the system's "learning budget" on the specific problem distribution at hand, trading generality for reliability within the target domain.

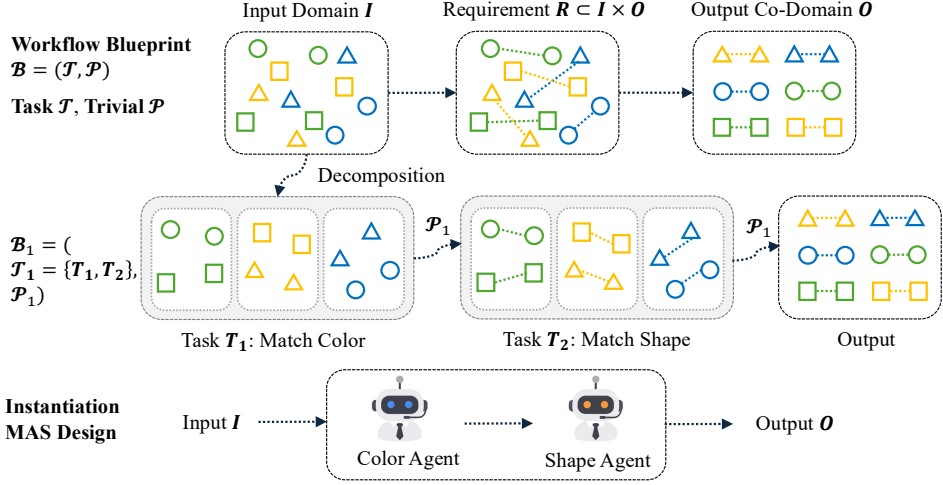

Figure 1: Illustration of the KtR strategy: heuristic, prior-guided decomposition of a complex task into sub-tasks, each instantiated as a coordinated LLM agent within a multi-agent architecture.

**Definition 3.1.** *A **well-formulated task** is a tuple $T = (I, O, R)$, consisting of*

- *Input domain $I$: an unambiguous description of all admissible inputs.*
- *Output co-domain $O$: an unambiguous description of all admissible outputs.*
- *Requirement relation $R \subset I \times R$: a relation such that for each input $x \in I$ it defines explicitly the subset $R(x) \subset O$ as the set of outputs that are considered correct.*

**Definition 3.2.** *A **workflow blueprint** $\mathcal{B} = (\mathcal{T}, \mathcal{P})$ consisting of*

- *A finite set of well-formulated tasks $\mathcal{T} = \{T_1, \cdots, T_n\}$.*
- *An orchestration protocol $\mathcal{P}$ that specifies: (i) The control-flow graph that determines when each $T_i$ is invoked. (ii) The data-dependency edges that map outputs of some tasks to inputs of others. (iii) Any global invariants, error-handling rules, or communication channels required to realize the end-to-end objective of $\mathcal{B}$.*

**Definition 3.3.** *Given a workflow blueprint $\mathcal{B} = (\mathcal{T}, \mathcal{P})$, a **decomposition** selects a task $T \in \mathcal{T}$ and replace it with a sub-blueprint $\mathcal{B}_T = (\mathcal{T}_T, \mathcal{P}_T)$ such that (1). Each task $T' \in \mathcal{T}_T$ is strictly simpler than $T$. (2). The composite protocol $\mathcal{P}'$, obtained by embedding $\mathcal{P}_T$ in place of $T$ inside $\mathcal{P}$, preserves all external interface of $T$. The result of the decomposition is a new blueprint $\mathcal{S}' = (\mathcal{T}', \mathcal{P}')$, where $\mathcal{T}' = (\mathcal{T} \backslash \{T\}) \cup \mathcal{T}_T$.*

**Definition 3.4.** *Let $\mathcal{M}$ be a set of LLM models. A well-formulated task $T$ is said to be $\mathcal{M}$-**tractable** if a model inside $\mathcal{M}$ satisfies the requirement relation $R_T$ with high, empirically verified accuracy, after optional augmentations (e.g., chain-of-thought prompting, tool calls, self-reflection loops, or fine-tuning).*

**Definition 3.5.** *Given a set of LLM models $\mathcal{M}$ and a blueprint $\mathcal{B}$, an $\mathcal{M}$-**tractable hierarchy** is a sequence of decompositions*

$$\mathcal{B} \overset{D_1}{\rightsquigarrow} \mathcal{B}_1 \overset{D_2}{\rightsquigarrow} \cdots \overset{D_n}{\rightsquigarrow} \mathcal{B}_n$$

*such that each task in the terminal blueprint $\mathcal{B}_n$ is $\mathcal{M}$-tractable in the sense of Definition 3.4.*

**Definition 3.6.** *Given a set of LLM models $\mathcal{M}$ and an $\mathcal{M}$-tractable blueprint $\mathcal{B} = (\mathcal{T}, \mathcal{P})$, a **system instantiation** is to instantiate $\mathcal{B}$ into a MAS in the following way.*

- *Create one agent $A_i$ per task $T_i \in \mathcal{T}$, bundling necessary augmentations with the agent.*
- *We implement the orchestration protocol $\mathcal{P}$ as message-passing or function calls among agents, preserving data-dependencies and control flow.*

---

**Algorithm 1** KtR Framework Pseudo-code

---

```
1: procedure KTR(T, M)
2:     B ← CREATEBLUEPRINT({T}, trivial_protocol)              # Start with the top-level task
3:     while exists U ∈ B.tasks and ¬MTRACTABLE(U, M) do
4:         U* ← CHOOSETASKTODECOMPOSE(U)                        # Select a non-tractable task
5:         B_sub ← DESIGNSUBBLUEPRINT(U*)                     # Define its sub-tasks and protocol
6:         B ← EMBEDSUBBLUEPRINT(B, U*, B_sub)             # Replace the task with its sub-blueprint
7:         ASSERTINTERFACEPRESERVED(B)                       # Ensure communications are valid
8:     end while
9:     MAS ← INSTANTIATESYSTEM()                                 # Begin building the MAS
10:    for all V ∈ B.tasks do
11:        aug ← SELECTAUGMENTATIONS(V, M)               # Select a cost-effective augmentation
12:        agent ← CREATEAGENT(V, M, aug)           # Create a specialized agent based on the definition
13:        MAS.AddAgent(agent)                             # Add the specialized agent to the system
14:    end for
15:    IMPLEMENTPROTOCOL(MAS, B.protocol)                  # Wire up agents based on the blueprint
16:    return MAS                                           # Return the final multi-agent system
17: end procedure
```

---

This three-step procedure—algorithmic blueprint, tractable hierarchy construction, and system instantiation—provides a principled pathway from a complex task to a deployable MAS solution with correctness hinges on model capabilities that have been explicitly validated. To demonstrate the practical application and efficacy of the KtR framework, we investigate two case studies. In each case, we use a well-understood algorithm to decompose the complex problem into an M-tractable hierarchy and instantiated MAS.

# 4 EXPERIMENT DESIGN

## 4.1 PROOF-OF-CONCEPT: 0/1 KNAPSACK PROBLEM (KSP)

To furnish a clear proof-of-concept for KtR, we start with the classical NP-hard Knapsack Problem (KSP), a staple in resource allocation, logistics, and investment planning. Using the lightweight, general-purpose GPT-4o-mini as the MAS backbone, we establish a modest baseline that allows us to highlight how KtR MAS choreography amplifies a small model's capability well beyond its limits.

**Problem Formulation**. For KSP, the input is a tuple $(\vec{w}, \vec{v}, W)$ where $\vec{w}$ and $\vec{v}$ are two $N$-dimensional vectors representing the weight and value of $n$ items, and $W$ is the weight capacity. Then the objective of the Knapsack problem can be formulated as finding the following (optimal) value:

$$Z = \max\{ \ \vec{x} \cdot \vec{v} \mid \vec{x} \in \{0,1\}^N, \ \vec{x} \cdot \vec{w} \leq W \ \}.$$

Here $\vec{x} \in \{0,1\}^N$ is the state vector, representing whether an item is chosen or not. This problem, where each item can either be fully included or not at all, is commonly known as the 0/1 KSP.

**Problem Solution** A classic approach to the Knapsack Problem iteratively enumerates all feasible states—a dynamic programming strategy (Bellman, 1957). Formulated in the form of mathematical induction, the initial state is $S_0 = \{(0,0)\}$, and we add items in inductively, with capacity being aware: for each $k$, assuming that $S_{k-1}$ has been obtained, then we form:

$$S_{add} = \{(w + w_k, v + v_k) \text{ for all } (w, v) \in S_{k-1}\}.$$

Then we trim by the capacity $S_{trimmed} = \{(w,v) \in S_{add} \mid w \leq W\}$ and union the two set of states to create $S_k$: $S_k = S_{k-1} \cup S_{trimmed}$. After running through all items and obtaining the set $S_N$, we pick the element in $S_N$ with maximal value, as the solution to the KSP. Specific details of this method is presented in Appendix C.

**KtR MAS Design**. Following the "Know the Ropes" heuristic, the iterative dynamic programming solution for the KSP is decomposed into tasks for three specialized agents, including: (i) **Worker Agent**: Computing the set $S_{add}$ from $S_{k-1}$ and the k-th item $(w_k, v_k)$.(ii) **Trimmer Agent**: Obtain $S_{trimmed}$ from $S_{add}$ and the capacity $W$. (iii) **Reporter Agent**: Find the element with maximal value within the final state set $S_N$. (iv) **System Controller**: The controller orchestrates the overall process. After initialization, it controls the loop on $k$. For each $k$, it sends $S_{k-1}$ and $(w_k, v_k)$ to Worker Agent, and then send the result plus $W$ to the Trimmer Agent. The Controller then takes the union to obtain $S_k$. Once all items are processed, the Controller invokes the Reporter Agent for final result. Specific prompts for these agent design are attached in Appendix E.

## 4.2 PROOF OF SCALABILITY—TASK ASSIGNMENT PROBLEM (TAP)

Building on the previous section where KtR already stretched the capabilities of the compact GPT-4o-mini on the Knapsack baseline, we now test the framework's scalability. We upgrade the backbone to the larger o3-mini and tackle the more demanding Task-Assignment Problem (TAP), demonstrating that the framework's performance rises in lockstep with the underlying model's capacity.

**Problem Formulation**. TAP seeks to optimally assign a set of $N$ workers to $N$ tasks, where each potential assignment incurs a specific cost to minimize the total cost. The input is an $N \times N$ matrix $C$ representing the cost. Then the objective of the TAP is to find the optimized value:

$$Z = \max_{\sigma \in \mathfrak{S}_N} \sum_{i=1}^{n} C_{i\sigma(i)}.$$

Here $\mathfrak{S}_N$ represents the set of all permutations of $N$ elements, representing arrangements that assign different tasks to different agents.

**Problem Solution**. The typical solution for TAP is using the Hungarian algorithm (Kuhn, 1955), which provides a polynomial-time method to find the objective value $Z$. We summaize the algorithm as follows. (i) **Step 1. Row Reduction.** For each row, we subtract each entry with the minimal value of all entries. This create a matrix $C'$. (ii) **Step 2. Column Reduction.** Same operation for each column to obtain a new matrix $C''$. (iii) **Step 3. Zero Covering.** We then find the smallest collection $\mathcal{C}$ of rows and columns to cover all zeroes. Let $L$ be the size. If $L = N$, then we skip Step 4. (iv) **Step 4. Matrix Improvement.** If $L < N$, we find the minimal value $m$ of all entries that are not

covered by the collection $\mathcal{C}$, and then subtract $m$ from all uncovered entries and add $m$ to all covered entries. Let $C'''$ be the resulting matrix. (v) **Step 5. Assignment Identification**. If $L = N$, then we attempt to find a collection of zeroes in $C''$ or $C'''$ in which no two are on the same row or column. The position of the zeroes represents the optimal task assignment. Specific details of this method is presented in Appendix C.

**KtR MAS Design**. We apply the KtR methodology to create the MAS. As explained in Section 5.2, based on test results of the agentic tasks and heuristic argument, we further decompose step 3 into two agents. For better presentation, let $N$ be the size of the original TAP problems, i.e., the number of tasks and resources in the problem. (i)**Row Reducer**: Realizes Step 1 and obtain the reduced matrix $C'$. (ii) **Column Reducer**: Realizes step 2 and obtain $C''$. (iii) **Matcher**: Find a maximal collection of zeroes in the reduced matrix $C''$ in which no two are on the same row or column. Let $L$ be the size of the collection. (iv) **Painter**: Find a minimal collection of rows and columns covering all zeroes when $L < n$. (v) **Normalizer**: Creates more zeroes outside of the selected rows and columns to get $C'''$. (iv) **Reporter**: Report the final answer when $L = N$. (vii) The **System Controller**: Arranges task for Row Reducer and Column Reducer linearly, then controls a loop: Matcher finds a set of zeroes and then Controller checks the size to determine whether to break the loop or not. In the loop, Painter is then called in to find the collection of lines and Normalizer follows to create more zeroes. Outside the loop, the Reporter is called to deduce the final answer. Specific prompts for these agent design are attached in Appendix E.

## 5 EXPERIMENT RESULT

Our experimental protocol unfolds in two stages. First, we run a uniform benchmark across a suite of baseline models—including several GPT and Llama variants—to fix a reference point for each task. The second stage then splits by objective: For KSP we deliver a proof of concept, while for TAP we provide a proof of scalability. For ground truth, we use python code to randomly generate problems, and then use the Google OR-Tools (Perron & Furnon, 2022) as in Appendix D to generate solutions to compare with.

### 5.1 EXPERIMENT RESULT FOR KSP

**Baseline LLM Performance** Figure 2 shows the baseline LLM performance across multiple difficulty levels. The accuracy across difficulty levels (from 3 to 8 items) in the KSP scenario reveals substantial performance variation among the tested LLMs. Among those, the GPT-o3-mini, as a reasoning model, consistently demonstrates superior accuracy. Model GPT-4.1 outperforming its smaller counterparts, namely GPT-4.1-mini and its primer GPT-4o-mini. Other LLMs, including Claude-haiku, Llama, and Qwen series also show performance degradation, with higher variability particularly evident at greater difficulty levels. Meanwhile, the performance

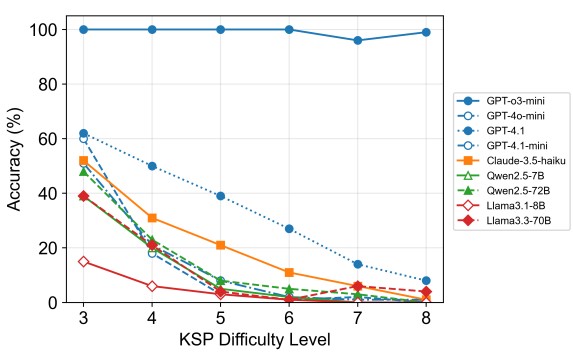

Figure 2: KSP baseline performance from single LLMs as well as the KtR MAS.

of final KtR MAS is also drawn in Figure 2. The comparison shows that KtR substantially boosts performance, validating its effectiveness.

**KtR MAS Performance** Based on Figure 2, GPT-4o-mini exhibits a pronounced performance decline beginning at instances of 4 items, underscoring its limited scalability to more complex scenarios; therefore, we select it as the backbone for our KtR framework design. Figure 3 further illustrates the resulting KtR MAS along with the experimental outcomes based on our strategy.

**(i) Single LLM performance**. We establish two baseline performances for GPT-4o-mini acting as a single agent to solve the KSP. First, the zero-shot GPT-4o-mini is directly prompted with KSP instances. As Figure 3B s shows, its accuracy decreases from 60% for 3 items to 0% (8 items).

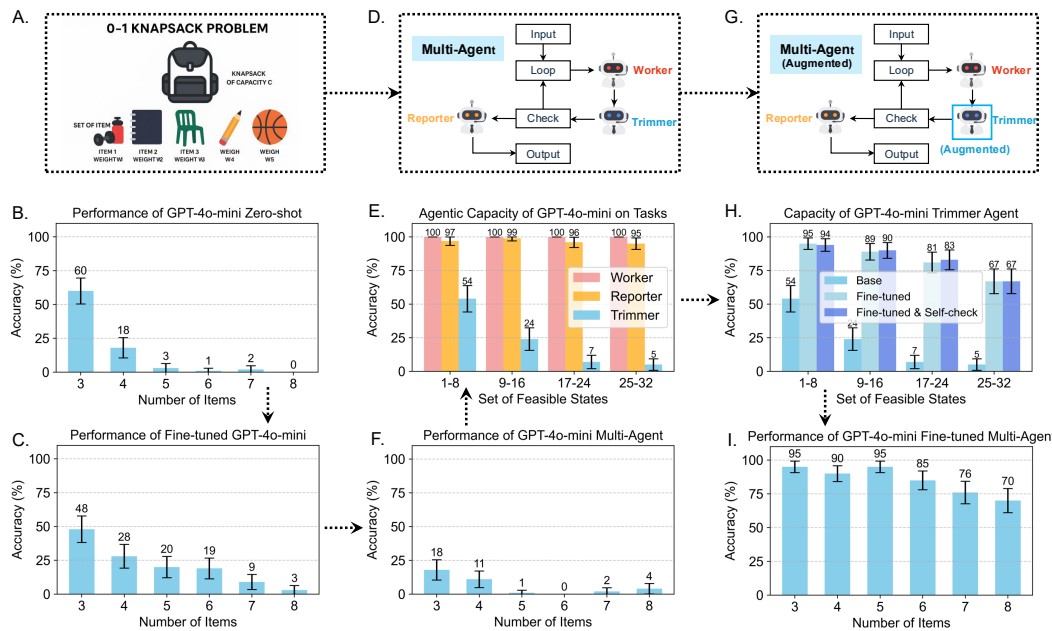

Figure 3: KSP evaluation of the KtR stategy. **B**: Zero-shot accuracy of the baseline model. **C**: Zero-shot accuracy after a light, task-specific fine-tune of the same model. **D** & **G**: Blueprints of the MAS without (**D**) and with (**G**) augmentations. **E**: Per-agent accuracies before augmentation, revealing the system's bottleneck. **H**: Boost delivered by two targeted augmentations—task-level fine-tuning and self-check prompting—applied to the bottleneck agent. **F** & **I**: Corresponding test accuracies for the two blueprints.

Second, we evaluate a fine-tuned GPT-4o-mini (standalone). Figure 3C indicates that fine-tuning offers some improvement over the zero-shot, but still as low as 3% for 8-item KSP.

**(ii) Standard MAS performance**. Following our KtR heuristic, we map the algorithm for KSP into a MAS design, illustrated in Figure 3F. Initially, each agent is driven by the standard, non-fine-tuned GPT-4o-mini. The performance of this standard MAS is presented in Figure 3F. Its performance descreases from 18% for 3 items to 4% for 8 items. This initial result implies that without augmenting the agents' abilities, the MAS does not effectively handle the task. We profile each agent in isolation (Figure 3E) and uncover a single choke point: Trimmer. Its accuracy collapses as the feasible-state set $S_k$ (cf. Section C.2) grows—54 % for 1–8 states, 24 % for 9–16, 7 % for 17–24, and just 5 % for 25–32. Because the algorithm loops once per state, even small per-iteration errors compound, and this cascading inaccuracy ultimately sinks the entire run.

**(iii) Augmented MAS performance**. To eliminate the bottleneck, we fine-tune the Trimmer's GPT-4o-mini backbone (Figure 3G, highlighted as 'Augmented Trimmer'). Accuracy leapt to 95 % for 1–8 feasible states, 89 % for 9–16, 81 % for 17–24, and 67 % for 25–32 (Figure 3H). Adding a lightweight self-check—prompting the model to audit its own answer—preserved or marginally improved these gains. Replacing the bottleneck with the fine-tuned Trimmer lifts end-to-end KSP accuracy to near-saturation across sizes (Figure 3I): 95 % for 3-item instances, 90 % for 4, 95 % for 5, 85 % for 6, 76 % for 7, and 70 % for 8. A single targeted upgrade thus turns KtR into a consistently high-performing solver as the problem scales.

## 5.2 EXPERIMENT RESULT ON TAP

**Baseline LLM Performance**  Figure 4 illustrates the baseline performance of multiple LLMs on the TAP task across multiple difficulty levels (from 3 to 8 tasks). The results reveal marked differences in model capabilities. The only reasoning model, GPT-4o-mini, consistently outperforms all others, exhibiting strong accuracy at lower difficulty levels, though its performance declines as task complexity increases. In contrast, GPT-4.1 demonstrates moderate but stable accuracy across all

difficulty levels, surpassing its mini-sized counterparts. Other models, including Claude-3.5-Haiku, Qwen2.5, and Llma-3 variants, show intermediate performance with variability.

We observe that single-agent models (e.g., GPT-3-mini, GPT-4-mini, GPT-4.1) drop to 30-50% accuracy at TAP levels 7-8, while KtR MAS maintains steady performance near 100%, even surpassing reasoning models, demonstrating its exceptional robustness and generalization capabilities.

**KtR MAS Performance** Based on Figure 4, GPT-o3-mini consistently outperforms other LLMs across all evaluated tasks, making it our choice for subsequent experiments. Our goal is to assess the scalability of our proposed strategy and investigate how its performance evolves as task difficulty increases. Figure 5 illustrates the MAS design and corresponding experimental outcomes obtained using our heuristic-based approach.

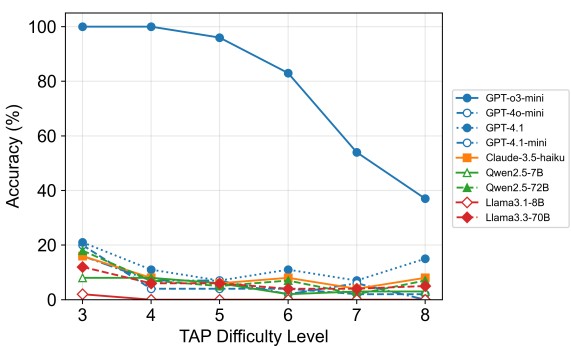

Figure 4: TAP baseline performance from single LLMs as well as the KtR mulit-agent system.

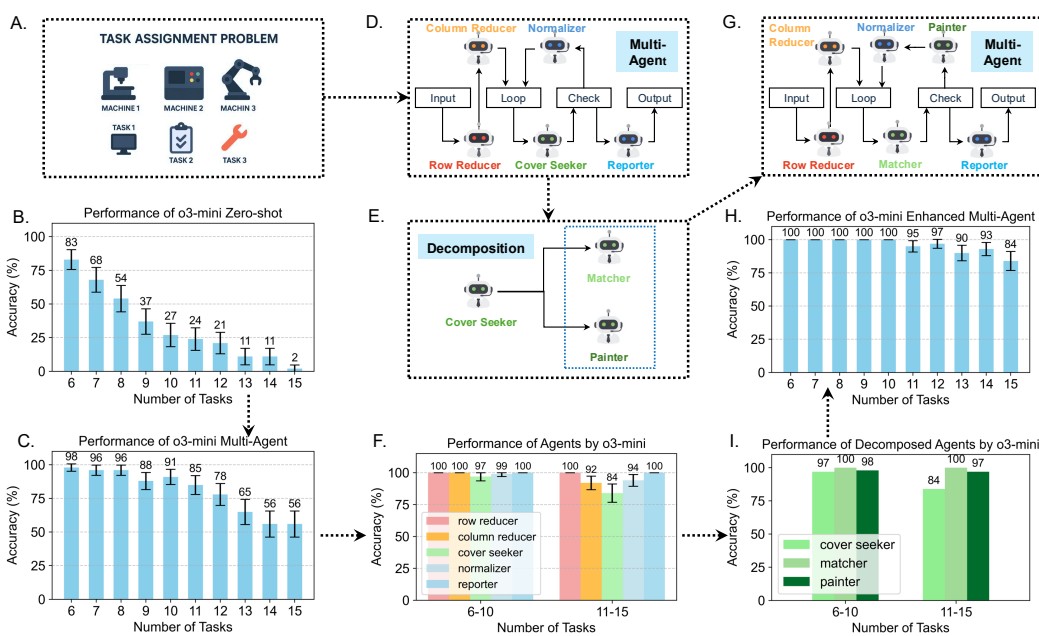

Figure 5: TAP evaluation of the KtR strategy. **B**: Zero-shot accuracy of the baseline model. **D**: Initial blueprint derived from the Hungarian algorithm; its end-to-end accuracy is shown in **C**. **F**: Per-agent accuracies within this blueprint, prompting the finer decomposition outlined in **E**. **I**: Side-by-side comparison of per-agent accuracies before and after decomposition. **G**: Final, decomposed blueprint, with overall accuracy presented in **H**.

**(i) Single LLM performance**. Again, we evaluate the baseline performance of using o3-mini as a single agent. The o3-mini model achieves a relatively high performance (83%) but decays quickly as in Figure 5B: 37% on problems of size 9, 21% on problems of size 12 and finally is reduced to 3% for problems of size 15.

**(ii) Standard MAS performance and further decomposition**. Guided by the Hungarian algorithm (Kuhn, 1955), our first KtR blueprint maps each step to a single agent; Step 3 from Section 4.2 relied on a lone Cover Seeker rather than the later "Matcher + Painter" pair. This baseline already scored 98 % (size 6), 88 % (size 9), 78 % (size 12), and 56 % (size 15), validating the approach.

We then stress-test each agent on two bands, with matrix sizes 6-10 and 11-15, to pinpoint weaknesses. One-shot agents are flawless: Row Reducer and Reporter reach 100 % on both bands, and Column Reducer hit 100 % / 92 %. Normalizer holds 99 % / 94 %, but Cover Seeker falls to 97 % / 84 %. Because Zero Seeker operates inside the main loop, its errors accumulate, making it the clear bottleneck for larger TAP instances.

We then perform a further decomposition of Step 3 in Section C.2 in a two-step process: Step 3.1. Finding a **maximal** collection of zero-entries, such that no two share a same row or column; Step 3.2. Finding a **minimal** collection of rows and columns covering all zero-entries. We believe this decomposition is helpful due to the following reasons. First, by a mathematical argument, the size of collections from sub-tasks 3.1 and 3.2 match. Second, a heuristic argument indicating that knowing the maximal collection of zeroes simplifies the task to find minimal collection of rows and columns. Last, the original Step 5 can then simply use the positions of the zeroes from Step 3.1, once optimal check passes. Note, this also explains why we prefer a further decomposition rather than fine-tuning the original agent, as a further decomposition improves the system flow as well. Empirical pays off, as shown in Figure 5I, Matcher reaches 100 % accuracy on both difficulty bands, while Painter climbs to 98 % and 97 %—a sharp jump from the original Cover Seeker's 97 % and 84 %.

**(iii) Augmented MAS performance**. Leveraging the refined decomposition, we deploy a six-agent system (Figure 5H) that solves size 6–10 instances almost flawlessly: almost 100 % accuracy versus 83 − 27 % for o3-mini zero-shot. It sustains high performance on size-11–15 tasks (95 %, 97 %, 90 %, 93 %, 84 %); even the dip at size 15 far exceeds the 3 % zero-shot baseline, highlighting the substantial capacity gain of our MAS.

# 6 DISCUSSION AND CONCLUSIONS

We present Know-the-Ropes (KtR), an engineering framework that turns algorithmic knowledge into reliable MAS via typed, verifiable blueprints. Our approach targets common failure modes in current MAS by enforcing structured decompositions, JSON-schema contracts, and local verification. Across evaluations, KtR shows effectiveness along complementary axes: KSP provides proof-of-concept by showing how a general-purpose small model (GPT-4o-mini) can handle complex reasoning tasks through structured decomposition, achieving 95% accuracy versus 3% zero-shot on our benchmark; TAP offers proof-of-scalability by leveraging the stronger reasoning capabilities of o3-mini on more demanding instances, reaching 100% accuracy up to size 10.

KtR's principled "**identify** → **improve** → **verify**" methodology indicates that disciplined decomposition plus targeted augmentation can substantially improve underperforming models on these tasks where both single-agent and unstructured MAS approaches struggle. The framework's effectiveness follows from domain-aligned inductive bias rather than prompt tinkering, broadly consistent with insights suggested by the No-Free-Lunch theorem. Our systematic process—bottleneck identification, task tractability assessment, and minimal augmentation—helps turn modest models into more reliable collaborators on decomposable problems without requiring ever-larger monoliths.

While our validation spans structured optimization and language tasks, future work should extend to less structured domains and address real-world complexities including noisy inputs and automated bottleneck detection. In addition, Although the per-agent inference cost is trending downward, we do not quantify absolute wall-clock latency, energy consumption, or controller overhead for large agent. Next, KtR positions algorithmic blueprints as a reliability-first complement to model scaling and prompt design, offering a systematic pathway from complex problems to deployable MAS solutions for decomposable tasks. As a limitation, our evaluation focuses on controlled, decomposable settings; broader studies are needed to assess robustness in open-ended workflows.

Finally, we offer practical guidance on when KtR pays off and how to evaluate it operationally. KtR is most effective when tasks admit natural subtask boundaries with local checks, stable typed interfaces, and controller overhead small relative to model calls. When objectives are globally entangled with sparse verification signals, a strong single agent or tool-augmented solver may be preferable. To support fair comparisons and blueprint refinement, we recommend reporting contract-level diagnostics alongside accuracy and cost: schema-conformance rate, controller rejection/repair rate, and the fraction of runs requiring controller intervention, so that gains reflect stronger structure rather than hidden prompt tuning.

## 7 ETHICS STATEMENT

This paper proposes Know-the-Ropes (KtR), a methodology for projecting algorithmic prior knowledge into typed, controller-mediated multi-agent system (MAS) blueprints, and evaluates it on synthetic combinatorial optimization tasks: the Knapsack Subset Problem (KSP) and the Task Assignment Problem (TAP). Our experiments do not involve human subjects, user data, or personally identifiable information. Problem instances are programmatically generated and ground-truth solutions are computed with Google OR-Tools (Apache 2.0), as described in Section D. Experiments use hosted large-language-model APIs (specifically GPT-4o-mini and o3-mini) for inference; we do not train or distribute any model weights. The scope of our study is limited to benign, synthetic tasks; we do not deploy agents in real-world settings. We are not aware of conflicts of interest related to this submission. An anonymous code repository is provided solely to facilitate inspection and reproduction and requires users to comply with the terms of the referenced third-party tooling.

## 8 REPRODUCIBILITY STATEMENT

We aim to make the study reproducible within the constraints of using hosted LLM APIs. The paper specifies the task settings, models, and orchestration details in text and figures (e.g., Figures illustrating the KSP and TAP blueprints). The data generation protocol and use of OR-Tools for ground truth are described in Section D. The exact prompt templates used for the zero-shot baselines and all agent roles are included at the end of the paper in Appendix E ("Prompt gallery"). An anonymous code base is available at https://anonymous.4open.science/r/KtR-codebase-5638, which is intended to help replicate the evaluation setup and run the reported experiments. For KSP, our MAS includes a fine-tuned component; the corresponding training sets are included in the code base. Reproducing our results requires access to the same API-served models (GPT-4o-mini and o3-mini); due to nondeterminism in API responses and service updates, results may exhibit small variance.

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

## A USE OF LLMs

During the development of this paper, we use transformer based large language models in the following aspects:

- Reference discovery: use the deep research tools from major providers to explore relevant work and literature.
- Code assistance: use coding agents to assist developing the code base of the current work.
- Grammar check: use LLMs to detect grammar errors in the drafty version of the paper, for better displaying our results.

## B APPENDIX: WEIGHTED NO FREE LUNCH THEOREM—WHY DOES AGENT DESIGN FAIL?

In this section, we present a weighted version of the No-Free-Lunch theorem. As the motivation, current approaches to MAS design can often result in overly general solutions that may exhibit suboptimal performance on specific and complex tasks. This sub-optimality arises partially from a lack of domain-specific inductive bias. To formalize this, we present a weighted variant of the No Free Lunch (NFL) theorem. The following demonstration, leveraging a weighted variant of the No Free Lunch theorem, quantitatively illustrates how inductive bias tailored to the target domain enhances performance. That is, We present a formal proof showing that, under a non-uniform prior concentrated on a problem-specific subset of functions, a specialized learning algorithm achieves strictly lower expected risk than a general-purpose algorithm.

Note the NLF theorem has been known to research community for more than two decades. Here what we present is a modification of the standard statement to better fit for our discussion on the MAS. As we didn't find in literature the precise version of the NFL theorem as we stated below, we also present a proof for self-containedness. We do not claim any originality of the theorem and the proof.

**Theorem B.1** (Weighted NFL). *Let $X$ be a finite input domain, $Y$ a finite label set, and $\mathcal{F} = Y^X$ the set of all functions $f: X \to Y$. Consider*

- *a general algorithm $A_0$ with constant expected loss $\varepsilon_0$ on every $f \in \mathcal{F}$,*
- *a specialized algorithm $A'$ satisfying*

$$L(h_{A'}, f) \leq \begin{cases} \varepsilon_1, & f \in \mathcal{F}', \\ \varepsilon_2, & f \notin \mathcal{F}', \end{cases}$$

*where $\varepsilon_1 < \varepsilon_0 < \varepsilon_2$, and*
- *a prior $P$ with $P(f \in \mathcal{F}') = p$ and $P(f \notin \mathcal{F}') = 1 - p$.*

*If*

$$p > \frac{\varepsilon_0 - \varepsilon_2}{\varepsilon_1 - \varepsilon_2},$$

*then the expected risk of $A'$ is strictly lower than that of $A_0$, i.e.*

$$R(A') < R(A_0).$$

*Proof.* By definition,

$$R(A_0) = \mathbb{E}_{f\sim P}[L(h_{A_0}, f)] = \varepsilon_0$$

$$R(A') = \mathbb{E}_{f\sim P}[L(h_{A'}, f)] = p\,\varepsilon_1 + (1-p)\,\varepsilon_2.$$

Hence

$$R(A') < R(A_0) \Longleftrightarrow p\,\varepsilon_1 + (1-p)\,\varepsilon_2 < \varepsilon_0$$
$$\Longleftrightarrow p(\varepsilon_1 - \varepsilon_2) > \varepsilon_0 - \varepsilon_2,$$

which rearranges to

$$p > \frac{\varepsilon_0 - \varepsilon_2}{\varepsilon_1 - \varepsilon_2}.$$

This completes the proof. $\qquad\square$

## C  APPENDIX: KSP AND TAP DESCRIPTION

In this appendix, we provide details about the KSP and TAP, including their problem description and algorithm based on which we design our MAS.

### C.1  KSP PROBLEM FORMULATION

The usual input of KSP involves a set of $N$ items, whose items are characterized by pairs $(w_i, v_i)$ of weights $w_i$ and values $v_i$, as well as a capacity value $W$. The goal of KSP is to find a subset of items such that the total weight does not exceed the given capacity while the total value is maximized. Mathematically, we record information of items by two vectors, both of dimension $N$: a weight vector $\vec{w} = (w_1, \cdots, w_N)$ and a value vector $\vec{v} = (v_1, \cdots, v_N)$. We also introduce the set of state-vectors $\{0, 1\}^N$, whose elements are vectors $\vec{x} = (x_1, \cdots, x_N)$ where entries $x_i$ takes values between 0 and 1, indicating whether an item is chosen in a subset or not:

$$x_i = \begin{cases} 1 & \text{item } i \text{ is chosen} \\ 0 & \text{item } i \text{ is excluded} \end{cases}$$

Thus state vectors controls which items is in the chosen subset, and the inner product of $\vec{x}$ with $\vec{w}$ and $\vec{v}$ then compute the total weight and total value for the given subset, respectively.

Given a weight vectors $\vec{w}$, a value vector $\vec{v}$, and the capacity constraint $W$, the objective of the Knapsack problem then can be formulated as finding the following (optimal) value:

$$Z = \max\{\ \vec{x} \cdot \vec{v} \mid \vec{x} \in \{0, 1\}^N,\ \vec{x} \cdot \vec{w} \leq W\ \}.$$

Here the maximal value is taken over all state vectors (or equivalently, all subsets of items) satisfying the constraint that the total weight $\vec{x} \cdot \vec{w}$ not exceeding the capacity $W$.

This version of the problem, where each item can either be fully included or not at all, is commonly known as the 0/1 KSP.

### C.2  KSP PROBLEM SOLUTION

A classic approach to the Knapsack Problem iteratively enumerates all feasible states—a dynamic-programming strategy first introduced by Bellman Bellman (1957). A feasible state can be defined by a pair $(current\_weight, current\_value)$ representing the accumulated weight and value of a set of items selected so far, such that $current\_weight \leq W$. We can describe the algorithm in the form of mathematical induction. We start with the initial set of feasible states $S_0 = \{(0, 0)\}$, representing an empty set of chosen items. We then add items in to form a set $S_k$ from $S_{k-1}$ inductively, with

capacity being aware: for each $k$, assuming that $S_{k-1}$ has been constructed, then we add the pair $(w_k, v_k)$ to all items in $S_{k-1}$ to form a new set $S_{add}$:

$$S_{add} = \{(w + w_k, v + v_k) \text{ for all } (w, v) \in S_{k-1}\}.$$

Then, we trim the set according to the capacity:

$$S_{trimmed} = \{(w, v) \in S_{add} \mid w \le W\}.$$

Note this also removes all repetitive states in the set. Finally we take the union of the two intermediate sets to create $S_k$:

$$S_k = S_{k-1} \cup S_{trimmed}.$$

The inductive step terminate when we have run through all items and obtaining the final set $S_N$, we pick the element in $S_N$ with maximal value, as the solution to the KSP. Explicitly,

$$Z = \max_{(w,v) \in S_N} v$$

## C.3 TAP PROBLEM FORMULATION

TAP seeks to optimally assign a set of $N$ resources (agents or workers) to $N$ tasks, where each potential assignment incurs a specific cost. With the constraint that each resource can only be assigned to one unique task, the objective of TAP is to find an assignment covering all tasks that minimizes the total cost. The resource-task specific cost is recorded in an $N \times N$ matrix $C$, where the entry $C_{ij}$ represents the cost associated with assigning resource $i$ to task $j$, for $i, j \in \{1, 2, \ldots, N\}$.

To formally define the problem, we introduce a set $\mathfrak{S}_n$ which can be described in either one of the following three equivalent ways:

- The group of automorphisms of the set $\underline{N} = \{1, 2, \cdots, N\}$.
- The set (or group) of bijections from the set $\underline{N}$ to itself.
- The set of all permutations involving $N$ elements.

Note elements in $\mathfrak{S}_N$ convey the idea that each resource is assigned to a unique task. Now, given the $N \times N$ cost matrix $C$, the objective of the TAP is to find the following (optimized) value

$$Z = \max_{\sigma \in \mathfrak{S}_N} \sum_{i=1}^n C_{i\sigma(i)}.$$

Note when we treat $\sigma \in \mathfrak{S}_N$ as a (bijective) map from $\underline{N} = \{1, 2, \cdots, N\}$ to itself, the notation $C_{i\sigma(i)}$ represents the entry on the $i$-th row and $\sigma(i)$-th column of the cost matrix $C$.

## C.4 TAP PROBLEM SOLUTION

The typical solution for TAP is using the Hungarian algorithm Kuhn (1955), which provides a polynomial-time method to find the objective value $Z$. We summarize the algorithm as follows.

**Step 1. Row Reduction.** For each row, we find the minimal element in the row and subtract it from all entries in the row, creating at least one zero on each row. Mathematically, starting from the original cost matrix $C_{N \times N}$, we create a new reduced matrix $C'$ such that for $i, j \in \{1, 2, \cdots, N\}$, we have

$$C'_{ij} = C_{ij} - \min_{1 \le k \le N} C_{ik}$$

**Step 2. Column Reduction.** Similarly, we further reduce $C'$ to $C''$ as follows: For each column, we find the minimal element in the column and subtract it from all entries in the column, guaranteeing at least one zero on each column. Mathematically, take

$$C''_{ij} = C'_{ij} - \min_{1 \le k \le N} C_{kj}$$

**Step 3. Find covering lines.** We then find a smallest collection of rows and columns to cover all zeroes. Here smallest is the sense of the number of elements in the collection (of rows and column), over all possible such collections. If the size of this minimal collection, denoted by $L$, coincides with $N$, the size of the problem, then we skip Step 4 to enter the final stage of the algorithm.

**Step 4. Matrix Improvement.** However, if $L < N$, we need an improvement for the matrix $C''$ before looping back to Step 3: given the minimal collection of rows and columns from Step 3, we find the minimal value of all entries that are not covered, and then subtract this minimal value from all uncovered entries of $C''$, and then add this minimal value to all entries of $C''$ that are covered twice, i.e., by both rows and columns. Let $C'''$ be the resulting matrix.

**Step 5. Assignment Identification**. Once the condition $L = N$ is met, the final step is to identify the optimal assignment. This involves selecting a set of $N$ independent zeros from the current matrix $C'''$, such that no two selected zeros share the same row or column. Each selected zero at position $(i, j)$ corresponds to assigning agent $i$ to task $j$. The total cost of this optimal assignment is then calculated by summing the costs from the original cost matrix $C$ corresponding to selected zero positions.

A non-trivial fact guaranteed by the Hungarian algorithm is that, in Step 5 the collection of zeroes might not be unique, while different collections are deemed to result in the same total summation of corresponding entries in the original cost matrix $C$.

## D  GROUND-TRUTH DATA PREPARATION

We utilize Google OR-Tools (Perron & Furnon, 2022) to generate optimal solutions—serving as ground-truth datasets—for both problem scenarios. OR-Tools is a widely adopted open-source software suite developed by Google for solving combinatorial optimization problems. It is renowned for its efficiency and reliability in addressing NP-hard challenges through advanced optimization algorithms. The suite is distributed under the permissive Apache License 2.0, allowing unrestricted use, modification, and distribution (Perron & Furnon, 2022).

For KSP, we generate random instances by assigning weights and values to items along with a maximum capacity constraint. Optimal solutions are then computed using OR-Tools' dynamic programming approach. For TAP, we similarly generate random cost matrices that represent the cost of assigning workers to tasks. Optimal assignments are obtained using the Hungarian algorithm as implemented in OR-Tools, which efficiently minimizes the total assignment cost.

## E  APPENDIX: PROMPT GALLERY

Note that all prompts we presented in the following, except for the self-check prompt for Trimmer Agent in KSP problem, are the system prompt for agents. The user prompt will only contain the precise problem to be handled by the agent in the form specified by the prompt.

### E.1  KSP PROMPTS

#### E.1.1  PROMPT FOR ZERO SHOT

```
You are an expert in the field of Knapsack Problem.

You are given a Knapsack Problem in the json format, of the
    following form:
{
    "id" : str,
    "items" : list of pairs of integers,
    "capacity" : int
}

Each pair in the list is a pair of integers of the form [weight,
    value], i.e., the first entry is the weight and the second
    entry is the value.

Your task is to solve the Knapsack Problem and provide the
    optimal solution. That is, you need to find a subset of the
    pairs that maximizes the total value, subject to the
```

constraint that the total weight of the subset is less than or equal to the capacity.

Please think step by step when solving the problem.

You need to return the optimal solution in the following json format:
{
    "max_value" : int,
}
Please only return the json format, nothing else.

### E.1.2   PROMPT FOR WORKER AGENT

You are a key member of a multi−agent team collaboratively solving the Knapsack Problem. Your specific role is the Worker, responsible for performing mathematical computations for the team.

You will receive input in the following JSON format:
{"c_list": [[int, int], ...], "s_item": [int, int]}
Each pair within 'c_list' contains two integers.

Your task is to:
− Add 's_item' to each pair in 'c_list' entry−wise. For instance, if a pair in 'c_list' is '[2, 5]' and 's_item' is '[3, 4]', the result should be '[2+3, 5+4] = [5, 9]'.

To ensure accuracy:
− Proceed systematically, applying step−by−step reasoning.
− Carefully perform each addition individually for all pairs provided in the list.

Your response must strictly follow this JSON format:
{"n_list": [[int, int], ...]}

Return only the specified JSON object without any additional commentary or text.

### E.1.3   PROMPT FOR TRIMMER AGENT

You are a key member of a multi−agent team collaboratively solving the Knapsack Problem. Your specific role is the Trimmer, responsible for trimming the list based on the given capacity constraint.

You will receive input in the following JSON format:
{"n_list": [[int, int], ...], "capacity": int}

Each pair within 'n_list' contains two integers: the first integer represents the weight, and the second integer represents the value.

Your task is to:
− Carefully analyze each pair in the provided list.
− Remove all pairs whose weight (the first integer) strictly exceeds the specified capacity.
− If identical pairs appear multiple times, retain only one instance of each.

To ensure accuracy:
- Proceed systematically, applying step-by-step reasoning.
- Verify each pair carefully against the capacity constraint.

Your response must strictly follow this JSON format:
{"t_list": [[int, int], ...]}

Return only the specified JSON object without any additional
    commentary or text.

### E.1.4    PROMPT FOR REPORTER AGENT

You are a key member of a multi-agent team collaboratively
    solving the Knapsack Problem. Your specific role is the
    Reporter, responsible for determining and clearly reporting
    the final answer based on the provided information.

You will receive input in the following JSON format:
{"c_list": [[int, int], ...]}
Each pair within 'c_list' contains two integers: the first
    integer represents the weight, and the second integer
    represents the value.

Your task is to carefully analyze this list, identify the pair
    with the maximal value (the second integer in each pair), and
    report only that maximal value. If the list is empty, then
    report the maximal value as 0.

To ensure accuracy:
- Proceed systematically, applying step-by-step reasoning.
- Carefully examine every pair in the provided list.

Your response must strictly follow this JSON format:
{"max_value": int}

Return only the JSON object as specified above, without any
    additional commentary or text.

### E.1.5    SELF-CHECK PROMPT FOR TRIMMER AGENT

To better fulfill your task, please conduct a double check on the
    result you just provided. If your answer is already correct,
    please confirm by copying the last output.

When double check, please pay attention to the following typical
    types of mistakes:

In particular, please check if you made any typical mistakes as
    listed below:
1. If you added in a pair that is not in the original n_list.
2. If there is still a pair in the t_list that still exceeds the
    capacity.
3. If there is a pair in n_list that does not exceed the capacity
    but is not in the t_list.

If you found any errors, please create a corrected answer.

In either case, please follow the format requirement of the
    output.

E.2  TAP PROMPTS

E.2.1  PROMPT FOR ZERO SHOT

You are an expert in solving the Assignment Problem. In the
    assignment problem, there are n workers and n jobs. Each
    worker has a cost of assigning to each job. Each worker can
    only be assigned to one job. Your task is to find the optimal
    assignment of workers to jobs that minimizes the total cost.

You are given the problem in the following json format:

{
    "id" : str,
    "cost_matrix" : list of lists of integers
}

The cost matrix is a square matrix of size n x n, where n is the
    number of workers and jobs, in the form of a nested list
    [[int, int, ...], [int, int, ...], ...]. The (i, j)th entry
    of the matrix represents the cost of assigning the ith worker
    to the jth job.

Your task is to find the optimal assignment of workers to jobs
    that minimizes the total cost.

Please think step by step when solving the problem.

You need to return the optimal assignment in the following json
    format:

{
    "optimal_cost" : int
}

Please only return the json format, nothing else.

E.2.2  PROMPT FOR ROW REDUCER AGENT

You are given a matrix in the following json format:

{
    "matrix" : list of lists of integers
}

The matrix is in the form of a nested list [[int, int, ...],
    [int, int, ...], ...].

Your task is to reduce the matrix by subtracting the minimum
    value of each row from all the elements in that row.

Please think step by step when solving the problem:
Step 0: Work on one row at a time.
Step 1: Find the minimum value of the row.
Step 2: Subtract the minimum value of the row from all the
    elements in that row.
Step 3: Return the reduced matrix in the following json format:

{"reduced_matrix" : list of lists of integers}

Please only return the json format, nothing else.

### E.2.3 PROMPT FOR COLUMN REDUCER AGENT

You are given a matrix in the following json format:

```
{
    "matrix" : list of lists of integers
}
```

The matrix is in the form of a nested list [[int, int, ...], [int, int, ...], ...].

Your task is to reduce the matrix by subtracting the minimum value of each column from all the elements in that column.

Please think step by step when solving the problem:
Step 0: Work on one column at a time.
Step 1: Find the minimum value of the column.
Step 2: Subtract the minimum value of the column from all the elements in that column.
Step 3: Return the reduced matrix in the following json format:

{"reduced_matrix" : list of lists of integers}

Please only return the json format, nothing else.

### E.2.4 PROMPT FOR ZERO SEEKER AGENT

You are given a problem in the following json format:

```
{
    "matrix" : list of lists of integers
}
```

The matrix is in the form of a nested list [[int, int, ...], [int, int, ...], ...].

Your task is to find a smallest collection of rows and columns of the matrix, such that any zeroes in the matrix is contained in a chosen row or column. Small means the sum of the sizes of the row and column collections is the smallest possible.

Please think step by step when solving the problem, and return your response in the following json format:

{"collum_collection" : [int, int, ...], "row_collection" : [int, int, ...]}

The integers in the collum_collection and row_collection are the indices of the rows and columns that you choose.

Please only return the json format, nothing else.

### E.2.5 PROMPT FOR MATCHER AGENT

You are given a matrix in the following json format:

```
{
    "matrix" : list of lists of integers
}
```

The matrix is in the form of a nested list [[int, int, ...], [int, int, ...], ...].

Your task is to find the largest collection of zeroes in the matrix, such that no two zeroes are in the same row or column.

Please think step by step when solving the problem, and return your response in the following json format:

{"largest_collection" : [[int, int], [int, int], ...]}

The list of pairs of integers is in the form of [[row_index, column_index], [row_index, column_index], ...].

Please only return the json format, nothing else.

### E.2.6 PROMPT FOR PAINTER AGENT

You are given a problem in the following json format:

```
{
    "matrix" : list of lists of integers
    "collection" : list of lists of integers
}
```

The matrix is in the form of a nested list [[int, int, ...], [int, int, ...], ...].

The collection is in the form of a nested list [[int, int], [int, int], ...].

Your task is to find a smallest collection of rows and columns of the matrix, such that any zeroes in the matrix is contained in a chosen row or column. Small means the sum of the sizes of the row and column collections is the smallest possible.

To assist you, you are provided with a collection of zeroes in the input json format. The collection contains the positions of a maximal collection of zeroes in the matrix, such that no two zeroes are in the same row or column.

Please use this collection of zeroes to find the rows and columns as desired. More precisely, you should first choose one row or column for each zero in the collection, such that the chosen rows and columns cover as much of the zeroes in the matrix as possible. Then add in more rows or columns if needed.

Please think step by step when solving the problem, and return your response in the following json format:

{"collum_collection" : [int, int, ...], "row_collection" : [int, int, ...]}
```

The integers in the collum_collection and row_collection are the indices of the rows and columns that you choose.

Please only return the json format, nothing else.

### E.2.7 PROMPT FOR NORMALIZER AGENT

You are given a problem in the following json format:

```
{
    "matrix" : list of lists of integers
    "collumn_collection" : list of integers
    "row_collection" : list of integers
}
```

The matrix is in the form of a nested list [[int, int, ...], [int, int, ...], ...].
The collumn_collection and row_collection are the indices of some selected rows and columns that covers all the zeroes in the matrix.

Your task is the following:
1. Find the minimal value in the matrix that is not covered by the selected rows and columns.
2. If this value is 0, return the original matrix.
3. If this value is not 0, subtract this value from all uncovered entries in the matrix.
4. For the entries that covered by both a selected row and a selected column, add this value to the entries.
5. For the entries that are covered by a selected row or column, but not both, do nothing.
6. Please return the updated matrix in the following json format:

```
{"normalized_matrix" : list of lists of integers}
```

Please only return the json format, nothing else.

### E.2.8 PROMPT FOR REPORTER AGENT

You are given a problem in the following json format:

```
{
    "matrix" : list of lists of integers
    "collection" : list of lists of integers
}
```

The matrix is in the form of a nested list [[int, int, ...], [int, int, ...], ...].
The collection contains a set of entries of the matrix in the form of [[row_index, column_index], [row_index, column_index], ...].

Your task is the following:
1. Sum up the values of all the entries in the collection.
2. Return the total value in the following json format:

```
{"total_value" : int}
```

Please only return the json format, nothing else.